# Neural Approximate Sufficient Statistics for Implicit Models

**Yanzhi Chen**[1]*, **Dinghuai Zhang**[2]*, **Michael U. Gutmann**[1], **Aaron Courville**[2], **Zhanxing Zhu**[3]
[1]The University of Edinburgh,  [2]MILA,  [3]Beijing Institute of Big Data Research

## Abstract

We consider the fundamental problem of how to automatically construct summary statistics for implicit generative models where the evaluation of the likelihood function is intractable but sampling data from the model is possible. The idea is to frame the task of constructing sufficient statistics as learning mutual information maximizing representations of the data with the help of deep neural networks. The infomax learning procedure does not need to estimate any density or density ratio. We apply our approach to both traditional approximate Bayesian computation and recent neural likelihood methods, boosting their performance on a range of tasks.

## 1 Introduction

Many data generating processes can be well-described by a parametric statistical model that can be easily simulated forward but does not possess an analytical likelihood function. These models are called implicit generative models (Diggle & Gratton, 1984) or simulator-based models (Lintusaari et al., 2017) and are widely used in science and engineering domains, including physics (Sjöstrand et al., 2008), genetics (Järvenpää et al., 2018), computer graphics (Mansinghka et al., 2013), robotics (Lopez-Guevara et al., 2017), finance (Bansal & Yaron, 2004), cosmology (Weyant et al., 2013), ecology (Wood, 2010) and epidemiology (Chinazzi et al., 2020). For example, the number of infected/healthy people in an outbreak could be well modelled by stochastic differential equations (SDE) simulated by Euler-Maruyama discretization but the likelihood function of a SDE is generally non-analytical. Directly inferring the parameters of these implicit models is often very challenging.

The techniques coined as *likelihood-free inference* open us a door for performing Bayesian inference in such circumstances. Likelihood-free inference needs to evaluate neither the likelihood function nor its derivatives. Rather, it only requires the ability to sample (*i.e.* simulate) data from the model. Early approaches in approximate Bayesian computation (ABC) perform likelihood-free inference by repeatedly simulating data from the model, and pick a small subset of the simulated data close to the observed data to build the posterior (Pritchard et al., 1999; Marjoram et al., 2003; Beaumont et al., 2009; Sisson et al., 2007). Recent advances make use of flexible neural density estimators to approximate either the intractable likelihood (Papamakarios et al., 2019) or directly the posterior (Papamakarios & Murray, 2016; Lueckmann et al., 2017; Greenberg et al., 2019).

Despite the algorithmic differences, a shared ingredient in likelihood-free inference methods is the choice of summary statistics. Well-chosen summary statistics have been proven crucial for the performance of likelihood-free inference methods (Blum et al., 2013; Fearnhead & Prangle, 2012; Sisson et al., 2018). Unfortunately, in practice it is often difficult to determine low-dimensional and informative summary statistic without domain knowledge from experts. In this work, we propose a novel deep neural network-based approach for automatic construction of summary statistics. Neural networks have been previously applied to learning summary statistics for likelihood-free inference (Jiang et al., 2017; Dinev & Gutmann, 2018; Alsing et al., 2018; Brehmer et al., 2020). Our approach is unique in that our learned statistics directly target *global sufficiency*. The main idea is to exploit the link between statistical sufficiency and information theory, and to formulate the task of learning sufficient statistic as the task of learning information-maximizing representations of data. We achieve this with distribution-free mutual information estimators or their proxies (Székely et al., 2014; Hjelm

---

*Equal contribution. Correspondence to Yanzhi Chen (`rhythm.cyz@gmail.com`) or Dinghuai Zhang (`dinghuai.zhang@mila.quebec`). Codes available at: https://github.com/cyz-ai/neural-approx-ss-lfi.

et al., 2018). Importantly, our statistics can be learned jointly with the posterior, resulting in fast learning where the two can refine each other iteratively. To sum up, our main contributions are:

- We propose a new neural approach to automatically extract *compact*, *near-sufficient* statistics from raw data. The approach removes the need for careful handcrafted design of summary statistics.
- With the proposed statistics, we develop two new likelihood-free inference methods namely SMC-ABC+ and SNL+. Experiments on tasks with various types of data demonstrate their effectiveness.

## 2 BACKGROUND

**Likelihood-free inference**. LFI considers the task of Bayesian inference when the likelihood function of the model is intractable but simulating (sampling) data from the model is possible:

$$\pi(\boldsymbol{\theta}|\mathbf{x}_o) \propto \pi(\boldsymbol{\theta}) \underbrace{p(\mathbf{x}_o|\boldsymbol{\theta})}_{?} \tag{1}$$

where $\mathbf{x}_o$ is the observed data, $\pi(\boldsymbol{\theta})$ is the prior over the model parameters $\boldsymbol{\theta}$, $p(\mathbf{x}_o|\boldsymbol{\theta})$ is the (possibly) non-analytical likelihood function and $\pi(\boldsymbol{\theta}|\mathbf{x}_o)$ is the posterior over $\boldsymbol{\theta}$. We assume that, while we do not have access to the exact likelihood, we can still sample (simulate) data from the model with a simulator: $\mathbf{x} \sim p(\mathbf{x}|\boldsymbol{\theta})$. The task is then to infer $\pi(\boldsymbol{\theta}|\mathbf{x}_o)$ given $\mathbf{x}_o$ and the sampled data: $\mathcal{D} = \{\boldsymbol{\theta}_i, \mathbf{x}_i\}_{i=1}^n$ where $\boldsymbol{\theta}_i \sim p(\boldsymbol{\theta}), \mathbf{x}_i \sim p(\mathbf{x}|\boldsymbol{\theta}_i)$. Note that $p(\boldsymbol{\theta})$ is not necessarily the prior $\pi(\boldsymbol{\theta})$.

**Curse of dimensionality**. Different likelihood-free inference algorithms might learn $\pi(\boldsymbol{\theta}|\mathbf{x}_o)$ in different ways, nevertheless most existing methods suffer from the curse of dimensionality. For example, traditional ABC methods use a small subset of $\mathcal{D}$ closest to $\mathbf{x}_o$ under some metric to build the posterior (Pritchard et al., 1999; Marjoram et al., 2003; Beaumont et al., 2009; Sisson et al., 2007), however in high-dimensional space measuring the distance sensibly is notoriously hard (Sorzano et al., 2014; Xie et al., 2017). On the other hand, recent advances (Papamakarios et al., 2019; Lueckmann et al., 2017; Papamakarios & Murray, 2016; Greenberg et al., 2019) utilize neural density estimators (NDE) to model the intractable likelihood or the posterior. Unfortunately, modeling high-dimensional distributions with NDE accurately is also known to be very difficult (Rippel & Adams, 2013; Van Oord et al., 2016), especially when the available training data is scarce.

Our interest here is not to design a new inference algorithm, but to find a low-dimensional statistic $\mathbf{s} = s(\mathbf{x})$ that is (Bayesian) sufficient:

$$\pi(\boldsymbol{\theta}|\mathbf{x}_o) \approx \pi(\boldsymbol{\theta}|\mathbf{s}_o) \propto \pi(\boldsymbol{\theta})p(\mathbf{s}_o|\boldsymbol{\theta}), \tag{2}$$

where $s : \mathcal{X} \to \mathcal{S}$ is a deterministic function also learned from $\mathcal{D}$. We conjecture that the learning of $s(\cdot)$ might be an easier task than direct density estimation. The resultant statistic $\mathbf{s}$ could then be applied to a wide range of likelihood-free inference algorithms as we will elaborate in Section 3.2.

## 3 METHODOLOGY

### 3.1 NEURAL SUFFICIENT STATISTICS

Our new deep neural network-based approach for automatic construction of near-sufficient statistics is based on the infomax principle, as illustrated by the following proposition (also see Figure 1):

**Proposition 1.** *Let $\boldsymbol{\theta} \sim p(\boldsymbol{\theta})$, $\mathbf{x} \sim p(\mathbf{x}|\boldsymbol{\theta})$, and $s : \mathcal{X} \to \mathcal{S}$ be a deterministic function. Then $\mathbf{s} = s(\mathbf{x})$ is a sufficient statistic for $p(\mathbf{x}|\boldsymbol{\theta})$ if and only if*

$$s = \underset{S:\mathcal{X}\to\mathcal{S}}{\arg\max} \ I(\boldsymbol{\theta}; S(\mathbf{x})),$$

*where $S$ is deterministic mapping and $I(\cdot; \cdot)$ is the mutual information between random variables.*

*Proof.* We defer the complete proof to the appendix. This proposition is a variant of Theorem 8 in (Shamir et al., 2010) with an adaption to the likelihood-free inference scenario. □

This important result suggests that we could find the sufficient statistic $\mathbf{s} = s(\mathbf{x})$ for a likelihood function $p(\mathbf{x}|\boldsymbol{\theta})$ by maximizing the mutual information (MI) $I(\boldsymbol{\theta}; \mathbf{s}) = KL[p(\boldsymbol{\theta}, \mathbf{s})\|p(\boldsymbol{\theta})p(\mathbf{s})]$ between $\boldsymbol{\theta}$ and $\mathbf{s}$. Moreover, as our interest is in maximizing MI rather than knowing its precise value,

Figure 1: *Left*. Traditional likelihood-free inference algorithm needs handcrafted design of summary statistic, which requires expert knowledge. *Right*. Our method automatically mines a low dimensional, near-sufficient statistic $\mathbf{s}$ of $\mathbf{x}$ via the infomax principle, which removes the need for careful summary statistic design. Furthermore, this statistics can be re-learned as the posterior inference proceeds.

we can maximize a non-KL surrogate, which may have an advantage in e.g. estimation accuracy or computational efficiency (Székely et al., 2014; Hjelm et al., 2018; Ozair et al., 2019). To this end, we utilize the following two non-KL estimators:

**Jensen-Shannon divergence** (JSD) (Hjelm et al., 2018): this non-KL estimator is shown to be more robust than KL-based ones. More specifically, it is defined as:

$$\hat{I}^{\text{JSD}}(\boldsymbol{\theta}; \mathbf{s}) = \sup_{T:\Theta \times \mathcal{S} \to \mathbb{R}} \mathbb{E}_{p(\boldsymbol{\theta}, \mathbf{s})} \left[ -\text{sp}(-T(\boldsymbol{\theta}, \mathbf{s})) \right] - \mathbb{E}_{p(\boldsymbol{\theta})p(\mathbf{s})} \left[ \text{sp}(T(\boldsymbol{\theta}, \mathbf{s})) \right], \tag{3}$$

where $\text{sp}(t) = \log(1 + \exp(t))$ is the softplus function. With this estimator, we set up the following objective for learning the sufficient statistics, which simultaneously estimates and maximizes the MI:

$$\max_{S,T} \ \mathcal{L}(S, T) = \mathbb{E}_{p(\boldsymbol{\theta}, \mathbf{x})} \left[ -\text{sp}\left( -T(\boldsymbol{\theta}, S(\mathbf{x})) \right) \right] - \mathbb{E}_{p(\boldsymbol{\theta})p(\mathbf{x})} \left[ \text{sp}\left( T(\boldsymbol{\theta}, S(\mathbf{x})) \right) \right], \tag{4}$$

where the two deterministic mappings $S$ and $T$ are parameterized by two neural networks. Note that we have used the law of the unconscious statistician (LOTUS) from equation 3 to equation 4. The mini-batch version of this objective is given in the appendix.

**Distance correlation** (DC) (Székely et al., 2014): unlike the JSD estimator, this estimator does not need to learn an additional network $T$, and can be learned much faster. It is defined as:

$$\hat{I}^{\text{DC}}(\boldsymbol{\theta}; \mathbf{s}) = \frac{\mathbb{E}_{p(\boldsymbol{\theta}, \mathbf{s})p(\boldsymbol{\theta}', \mathbf{s}')}[h(\boldsymbol{\theta}, \boldsymbol{\theta}')h(\mathbf{s}, \mathbf{s}')]}{\sqrt{\mathbb{E}_{p(\boldsymbol{\theta})p(\boldsymbol{\theta}')}[h^2(\boldsymbol{\theta}, \boldsymbol{\theta}')]} \cdot \sqrt{\mathbb{E}_{p(\mathbf{s})p(\mathbf{s}')}[h^2(\mathbf{s}, \mathbf{s}')]}}, \tag{5}$$

where $h(\mathbf{a}, \mathbf{b}) = \|\mathbf{a} - \mathbf{b}\| - \mathbb{E}_{p(\mathbf{b}')}[\|\mathbf{a} - \mathbf{b}'\|] - \mathbb{E}_{p(\mathbf{a}')}[\|\mathbf{a}' - \mathbf{b}\|] + \mathbb{E}_{p(\mathbf{a}')p(\mathbf{b}')}[\|\mathbf{a}' - \mathbf{b}'\|]$. Similar to the case of the JSD estimator, we set up the following objective for learning the sufficient statistics:

$$\max_{S} \ \mathcal{L}(S) = \frac{\mathbb{E}_{p(\boldsymbol{\theta}, \mathbf{x})p(\boldsymbol{\theta}', \mathbf{x}')}[h(\boldsymbol{\theta}, \boldsymbol{\theta}')h(S(\mathbf{x}), S(\mathbf{x}'))]}{\sqrt{\mathbb{E}_{p(\boldsymbol{\theta})p(\boldsymbol{\theta}')}[h^2(\boldsymbol{\theta}, \boldsymbol{\theta}')]} \cdot \sqrt{\mathbb{E}_{p(\mathbf{x})p(\mathbf{x}')}[h^2(S(\mathbf{x}), S(\mathbf{x}'))]}}, \tag{6}$$

where the deterministic mapping $S$ is parameterized by a neural network. Again LOTUS is used from equation 5 to equation 6. The mini-batch version of this objective is given in the appendix.

A comparison between the accuracy and efficiency of these two MI estimators (as well as other estimators (Belghazi et al., 2018; Ozair et al., 2019)) for infomax statistics learning is in the appendix.

With enough training samples and powerful neural networks, we can obtain near-sufficient statistics with either $s = \arg\max_S \max_T \mathcal{L}(S, T)$ or $s = \arg\max_S \mathcal{L}(S)$, depending on the estimator. The statistic $\mathbf{s}$ of data $\mathbf{x}$ is then given by

$$\mathbf{s} = s(\mathbf{x}). \tag{7}$$

In the above construction, we have not specified the form of the networks $S$ and $T$. For $S$, any prior knowledge about the data $\mathbf{x}$ could in principle be incorporated into its design. For example, for sequential data we can realize $S$ as a transformer (Vaswani et al., 2017), and for exchangeable data we can realize $S$ as a exchangeable neural network (Chan et al., 2018). Here we simply adopt a fully-connected architecture for $S$, and leave the problem-specific design of $S$ as future work. For $T$, we choose it to be a split architecture $T(\boldsymbol{\theta}, S(\mathbf{x})) = T'(H(\boldsymbol{\theta}), S(\mathbf{x}))$ where $T'(\cdot, \cdot), H(\cdot)$ are both MLPs. Therefore we separately learn low-dimensional representations for $\mathbf{x}$ and $\boldsymbol{\theta}$ before processing

them together. This could be seen as that we incorporate the inductive bias into the design of the networks that $\mathbf{x}$ and $\boldsymbol{\theta}$ should *not* interact with each other directly, based on their true relationship (for example, consider the likelihood function of exponential family: $L(\boldsymbol{\theta}; \mathbf{x}) \propto \exp(H(\boldsymbol{\theta})^\top S(\mathbf{x}))$).

We are left with the problem of how to select $d$, the dimensionality of the sufficient statistics. The Pitman-Koopman-Darmois theorem (Koopman, 1936) tells us that sufficient statistics with fixed dimensionality only exists for exponential family, so there is no universal way to select $d$. Here, we propose to use the following simple heuristics to determine $d$:

$$d = 2K \tag{8}$$

where $K$ is the dimensionality of $\boldsymbol{\theta}$ (which typically satisfies $K \ll D$). The rationale behind this heuristics is that the dimensionality of the sufficient statistics in the exponential family is $K$, and exponential family has been proven reasonably accurate for posterior approximation (see e.g. Thomas et al., 2021; Pacchiardi & Dutta, 2020). By doubling the dimensionality of the statistics to $2K$ we are likely to have a better representative power than the exponential family while still keeping $d$ small.

Furthermore, we have the following proposition comparing our method to the existing *posterior-mean-as-statistic* approaches (Fearnhead & Prangle, 2012; Jiang et al., 2017).

**Proposition 2.** *Let $\boldsymbol{\theta} \sim p(\boldsymbol{\theta})$ and $\mathbf{x} \sim p(\mathbf{x}|\boldsymbol{\theta})$. Let $s(\cdot)$ be a deterministic function that satisfies*

$$s = \underset{S:\mathcal{X}\to\mathcal{S}}{\arg\min} \ \mathbb{E}_{p(\boldsymbol{\theta},\mathbf{x})}[\|S(\mathbf{x}) - \boldsymbol{\theta}\|_2^2],$$

*then $\mathbf{s} = s(\mathbf{x})$ is generally not a maximizer of $I(S(\mathbf{x}); \boldsymbol{\theta})$ and hence it is not a sufficient statistic.*

*Proof.* We defer the proof to the appendix. $\square$

This proposition tells us that unlike our method, the existing (posterior-)mean-as-statistic approaches widely used in likelihood-free inference community lose information about the posterior, and it is only optimal for predicting the posterior mean (Fearnhead & Prangle, 2012; Jiang et al., 2017). When using this statistics in inference, it may yield inaccurate estimates of e.g. the posterior uncertainty. Nonetheless which statistics to use depends on the task, e.g. full posterior vs. point estimation.

## 3.2 DYNAMIC STATISTICS-POSTERIOR LEARNING

The above neural sufficient statistic could, in principle, be learned via a pilot run before the inference starts, as, for example, done in the work by Drovandi et al. (2011); Fearnhead & Prangle (2012); Jiang et al. (2017). Such a strategy requires extra simulation cost, and the learned statistic is kept fixed during inference. We propose a dynamic learning strategy below to overcome these limitations.

Our idea is to *jointly learn the statistic and the posterior in multiple rounds*. More concretely, at round $j$, we use the current statistic $s(\cdot)$ to build the $j$-th estimate to the posterior: $q_j(\boldsymbol{\theta}|\mathbf{s}_o) \approx \pi(\boldsymbol{\theta}|\mathbf{x}_o)$, and at round $j+1$, this estimate is used as the new proposal distribution to simulate data: $p_{j+1}(\boldsymbol{\theta}) \leftarrow q_j(\boldsymbol{\theta}|\mathbf{s}_o), \boldsymbol{\theta}_i \sim p_{j+1}(\boldsymbol{\theta}), \mathbf{x}_i \sim p(\mathbf{x}|\boldsymbol{\theta}_i)$. We then re-learn $s(\cdot)$ and $q(\cdot)$ with all the data up to the new round. In this process, $s(\cdot)$ and $q(\cdot)$ refine each other: a good $s(\cdot)$ helps to learn $q(\cdot)$ more accurately, whereas an improved $q(\cdot)$ as a better proposal in turn helps to learn $s(\cdot)$ more efficiently.

The theoretical basis of this multi-rounds strategy is provided by Proposition 1, which tells us that the sufficiency of the learned statistics is *insensitive* to the choice of $p(\boldsymbol{\theta})$, the marginal distribution of $\boldsymbol{\theta}$ in sampled data $\mathcal{D} = \{\mathbf{x}_i, \boldsymbol{\theta}_i\}_{i=1}^{nj}$. This means that we are indeed safe to use any proposal distribution $p_l(\boldsymbol{\theta})$ at any round $l$ in multi-rounds learning, and in such case $p(\boldsymbol{\theta})$ after round $j$ will be a mixture distribution formed by the proposal distributions of the previous rounds, i.e. $p(\boldsymbol{\theta}) = \frac{1}{j}\sum_{l=1}^{j} p_l(\boldsymbol{\theta})$.

---

**Algorithm 1** SMC-ABC+

> **Input:** prior $\pi(\boldsymbol{\theta})$, observed data $\mathbf{x}_o$
> **Output:** estimated posterior $\hat{\pi}(\boldsymbol{\theta}|\mathbf{x}^o)$
> **Initialization:** $\mathcal{D} = \varnothing, p_1(\boldsymbol{\theta}) = \pi(\boldsymbol{\theta})$
> **for** $j$ in $1$ to $r$ **do**
>   **repeat**
>     sample $\boldsymbol{\theta}_i \sim p_j(\boldsymbol{\theta})$ ;
>     simulate $\mathbf{x}_i \sim p(\mathbf{x}|\boldsymbol{\theta}_i)$ ;
>   **until** $n$ samples
>   $\mathcal{D} \leftarrow \mathcal{D} \cup \{\boldsymbol{\theta}_i, \mathbf{x}_i\}_{i=1}^n$
>   fit statistic net $s(\cdot)$ with $\mathcal{D}$ by equation 4 ;
>   sort $\mathcal{D}$ according to $\|s(\mathbf{x}_i) - s(\mathbf{x}_o)\|$ ;
>   fit $p(\boldsymbol{\theta}|\mathbf{s}_o)$ with the top $m$ $\boldsymbol{\theta}$s in $\mathcal{D}$;
>   $q_j(\boldsymbol{\theta}|\mathbf{s}_o) \propto p(\boldsymbol{\theta}|\mathbf{s}_o)\pi(\boldsymbol{\theta})/\sum_l^j p_l(\boldsymbol{\theta})$;
>   $p_{j+1}(\boldsymbol{\theta}) \leftarrow q_j(\boldsymbol{\theta}|\mathbf{s}_o)$;
> **end for**
> **return** $\hat{\pi}(\boldsymbol{\theta}|\mathbf{x}_o) = q_r(\boldsymbol{\theta}|\mathbf{s}_o)$

**Algorithm 2** SNL+

> **Input:** prior $\pi(\boldsymbol{\theta})$, observed data $\mathbf{x}_o$
> **Output:** estimated posterior $\hat{\pi}(\boldsymbol{\theta}|\mathbf{x}^o)$
> **Initialization:** $\mathcal{D} = \varnothing, p_1(\boldsymbol{\theta}) = \pi(\boldsymbol{\theta})$
> **for** $j$ in $1$ to $r$ **do**
>   **repeat**
>     sample $\boldsymbol{\theta}_i \sim p_j(\boldsymbol{\theta})$ ;
>     simulate $\mathbf{x}_i \sim p(\mathbf{x}|\boldsymbol{\theta}_i)$ ;
>   **until** $n$ samples
>   $\mathcal{D} \leftarrow \mathcal{D} \cup \{\boldsymbol{\theta}_i, \mathbf{x}_i\}_{i=1}^n$
>   fit statistic net $s(\cdot)$ with $\mathcal{D}$ by equation 4;
>   convert $\mathcal{D}$ with the learned $s(\cdot)$;
>   fit $q(\mathbf{s}|\boldsymbol{\theta})$ with converted $\mathcal{D}$ by equation 11;
>   $q_j(\boldsymbol{\theta}|\mathbf{s}_o) \propto \pi(\boldsymbol{\theta}) \cdot q(\mathbf{s}_o|\boldsymbol{\theta})$;
>   $p_{j+1}(\boldsymbol{\theta}) \leftarrow q_j(\boldsymbol{\theta}|\mathbf{s}_o)$;
> **end for**
> **return** $\hat{\pi}(\boldsymbol{\theta}|\mathbf{x}_o) = q_r(\boldsymbol{\theta}|\mathbf{s}_o)$

---

In practice, any likelihood-free inference algorithm that learns the posterior sequentially naturally fits well within the above joint statistic-posterior learning strategy. Here we study two such instances:

**Sequential Monte Carlo ABC (SMC-ABC)** (Beaumont et al., 2009). This classical algorithm learns the posterior in a non-parametric way within multiple rounds. Here, we consider a variant of it to better make use of the above neural sufficient statistic, and to re-use all previous simulated data. The new SMC-ABC algorithm estimates the posterior $q_j(\boldsymbol{\theta}|\mathbf{s}_o)$ at the $j$-th round as follows. We first sort data in $\mathcal{D} = \{\mathbf{x}_i, \boldsymbol{\theta}_i\}_{i=1}^{nj}$ according to the distances $\|s(\mathbf{x}_i) - s(\mathbf{x}_o)\|$. We then pick the top-$m$ $\boldsymbol{\theta}$s whose corresponding distances are the smallest. The picked $\boldsymbol{\theta}$s then follow $\boldsymbol{\theta} \sim p(\boldsymbol{\theta} \mid \mathbf{s}_o)$ as below:

$$p(\boldsymbol{\theta} \mid \mathbf{s}_o) \propto \sum_{l=1}^j p_l(\boldsymbol{\theta}) \cdot \Pr(\|\mathbf{s} - \mathbf{s}_o\| < \epsilon \mid \boldsymbol{\theta}), \tag{9}$$

where the threshold $\epsilon$ is implicitly defined by the ratio $\frac{m}{nj}$ (which automatically goes to zero as $j \to \infty$). We then fit $p(\boldsymbol{\theta}|\mathbf{s}_o)$ with the collected $\boldsymbol{\theta}$s by a flexible parametric model (e.g. a Gaussian copula), with which we can obtain the $j$-th estimate to the posterior by importance (re-)weighting:

$$q_j(\boldsymbol{\theta} \mid \mathbf{s}_o) \propto p(\boldsymbol{\theta} \mid \mathbf{s}_o)\pi(\boldsymbol{\theta})/\sum_{l=1}^j p_l(\boldsymbol{\theta}). \tag{10}$$

The whole procedure of this new inference algorithm, SMC-ABC+, is summarized in Algorithm 1.

**Sequential Neural Likelihood (SNL)** (Papamakarios et al., 2019). This recent algorithm learns the posterior in a parametric way, also in multiple rounds. The original SNL method approximates the likelihood function $p(\mathbf{x}|\boldsymbol{\theta})$ by a conditional neural density estimator $q(\mathbf{x}|\boldsymbol{\theta})$, which could be difficult to learn if the dimensionality of $\mathbf{x}$ is high. Here, we alleviate such difficulty with our neural statistic. The new SNL algorithm estimates the posterior $q_j(\boldsymbol{\theta}|\mathbf{s}_o)$ at the $j$-th round as follows. At round $j$, where we have $nj$ simulated data $\mathcal{D} = \{\boldsymbol{\theta}_i, \mathbf{x}_i\}_{i=1}^{nj}$, we fit a neural density estimator $q(\mathbf{s}|\boldsymbol{\theta})$ as:

$$q(\mathbf{s} \mid \boldsymbol{\theta}) = \arg\max_Q \sum_{i=1}^{nj} \log Q(s(\mathbf{x}_i) \mid \boldsymbol{\theta}_i), \tag{11}$$

where $s(\cdot)$ is the current statistic network and $Q$ is a neural density estimator (e.g. Durkan et al. (2019); Papamakarios et al. (2017)). With $nj$ being moderately large and $Q$ flexible enough, this would yield us $q(\mathbf{s}|\boldsymbol{\theta}) \approx p(\mathbf{s}|\boldsymbol{\theta})$. We then obtain the $j$-th estimate of the posterior by Bayes rule:

$$q_j(\boldsymbol{\theta} \mid \mathbf{s}^o) \propto \pi(\boldsymbol{\theta}) \cdot q(\mathbf{s}^o \mid \boldsymbol{\theta}). \tag{12}$$

The whole procedure of this new SNL algorithm, denoted as SNL+, is summarized in Algorithm 2.

## 4    RELATED WORKS

**Approximate Bayesian computation**. ABC denotes techniques for likelihood-free inference which work by repeatedly simulating data from the model and picking those similar to the observed data to estimate the posterior (Sisson et al., 2018). Naive ABC performs simulation with the prior, whereas MCMC-ABC (Marjoram et al., 2003; Meeds et al., 2015) and SMC-ABC (Beaumont et al., 2009; Sisson et al., 2007) use informed proposals, and more advanced methods employ experimental design or active learning to accelerate the inference (Gutmann & Corander, 2016; Järvenpää et al., 2019). To measure the similarity to the observed data, it is often wise to use low-dimensional summary statistics rather than the raw data. Here we develop a way to learn compact sufficient statistics for ABC.

**Neural density estimator-based inference**.  Apart from ABC, a recent line of research uses a conditional neural density estimator to (sequentially) learn the intractable likelihood (e.g SNL Papamakarios et al. (2019); Lueckmann et al. (2019)) or directly the posterior (e.g SNPE Papamakarios & Murray (2016); Lueckmann et al. (2017); Greenberg et al. (2019)). Likelihood-targeting approaches have the advantage that they could readily make use of any proposal distribution in sequential learning, but they rely on low-dimensional, well-chosen summary statistic. Posterior-targeting methods on the contrary need no design of summary statistic, but they require non-trivial efforts to facilitate sequential learning. Our approach (e.g SNL+) can be seen as taking the advantages from both worlds.

**Automatic construction of summary statistics**. A set of works have been proposed to automatically construct low-dimensional summary statistics. Two lines of them are most related to our approach. The first line (Fearnhead & Prangle, 2012; Jiang et al., 2017; Chan et al., 2018; Wiqvist et al., 2019; Dinev & Gutmann, 2018) train a neural network to predict the posterior mean and use this prediction as the summary statistic. These mean-as-statistic approaches, as analyzed previously in Proposition 2, indeed do not guarantee sufficiency. Rather than taking the predicted mean, the works (Alsing et al., 2018; Brehmer et al., 2020) take the score function $\nabla_{\boldsymbol{\theta}} \log p(\mathbf{x}|\boldsymbol{\theta})|_{\boldsymbol{\theta}=\boldsymbol{\theta}^*}$ around some fiducial parameter $\boldsymbol{\theta}^*$ as the summary statistic. However, these score-as-statistic methods are only *locally* sufficient around $\boldsymbol{\theta}^*$. Our approach differs from all these methods as it is *globally* sufficient for all $\boldsymbol{\theta}$.

**MI and ratio estimation**. It has been shown in the literature that many variational MI estimators $I(X;Y)$ also estimate the ratio $p(X,Y)/p(X)p(Y)$ up to a constant (Nowozin et al., 2016; Nguyen et al., 2010). Therefore our MI-based statistic learning method is closely related to ratio estimating approaches to posterior inference (Hermans et al., 2020; Thomas et al., 2021). The differences are 1) we decouple the task of statistics learning from the task of density estimation for LFI, which grants us the privilege to use any infomax representation learning methods that are ratio-free (Székely et al., 2014; Ozair et al., 2019); and 2) even if we do estimate the ratio, we do this in the low-dimensional space based on a sufficient statistics perspective, which is typically easier than in the original space.

## 5    EXPERIMENTS

### 5.1    SETUP

**Baselines**. We apply the proposed statistics to two aforementioned likelihood-free inference methods: (i) SMC-ABC (Beaumont et al., 2009) and (ii) SNL (Papamakarios et al., 2019). We compare the performance of the algorithms augmented with our neural statistics (dubbed as SMC-ABC+ and SNL+) to their original versions as well as the versions based on expert-designed statistics (details presented later; we call the corresponding methods SMC-ABC' and SNL'). We also compare to the sequential neural posterior estimate (SNPE) method[1] which needs no statistic design, as well as the sequential ratio estimate (SRE) method (Hermans et al., 2020) which is closely related to our MI-based method[2]. All methods are run for 10 rounds with 1,000 simulations each. The results presented below are for the JSD estimator; the DC estimator achieves similar accuracy (see appendix).

**Evaluation metric**. To assess the quality of the estimated posterior, we compare the Jensen-Shannon divergence (JSD) between the approximate posterior $Q$ and the true posterior $P$ for each method

---

[1]More specifically, the version B. We compare with SNPE-B (Lueckmann et al., 2017) rather than the more recent SNPE-C (Greenberg et al., 2019) due to its similarity to SRE shown in (Durkan et al., 2020).

[2]For fair comparison, we control that the neural network in SRE has a similar number of parameters/same optimizer settings as in our method. See the appendix for more details about the settings of the neural networks.

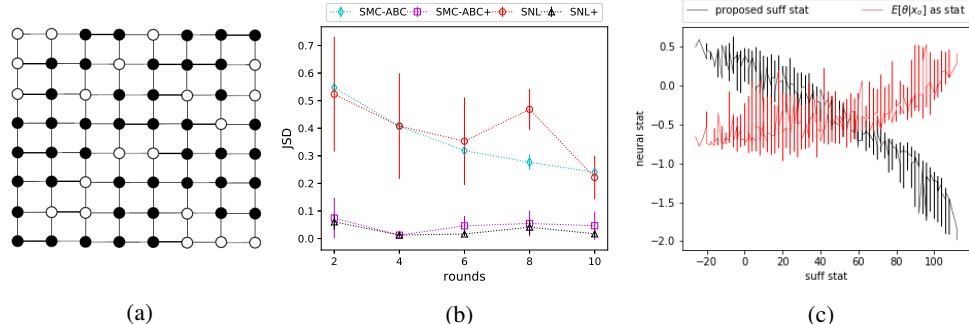

|  | (a) |  | (b) |  | (c) |

Figure 2: **Ising model**. (a) The 64D observed data $\mathbf{x}_o \in \{-1, 1\}^{64}$. (b) The JSD between the true and the learned posteriors. (c) The relationship between the learned statistics and the sufficient statistic.

| SMC' | SMC+ | SNL' | SNL+ | SRE | SNPE |
|---|---|---|---|---|---|
| $0.008 \pm 0.006$ | $0.046 \pm 0.051$ | $0.007 \pm 0.002$ | $0.015 \pm 0.011$ | $0.083 \pm 0.029$ | $0.058 \pm 0.039$ |

Table 1: **Ising model**. The JSD between the learned and true posterior with 10,000 simulations. Here SMC' and SNL' utilize the ground-truth sufficient statistics guided by human prior knowledge.

(see appendix). For the problems we consider, the true posterior $P$ is either analytically available, or can be accurately approximated by a standard rejection ABC algorithm (Pritchard et al., 1999) with known low-dimensional sufficient statistic (e.g $s(\mathbf{x}) \in \mathbb{Z}$) and extensive simulations (e.g $10^6$).

## 5.2 RESULTS

We demonstrate the effectiveness of our method on three models: (a) an Ising model; (b) a Gaussian copula model; (c) an Ornstein-Uhlenbeck process. The Ising model does not have an analytical likelihood but the posterior can be approximated accurately by rejection ABC due to the existence of low-dimensional, discrete sufficient statistic. The last two models have analytical likelihoods and hence analytical posteriors. These models cover the cases of graph data, i.i.d data and sequence data.[3]

**Ising model**. The first model we consider is a mathematical model in statistical physics that describes the states of atomic spins on a $8 \times 8$ lattice (see Figure 1(a)). Each spin has two states described by a discrete random variable $x_i \in \{-1, +1\}$, and is only allowed to interact with its neighbour. Given parameters $\boldsymbol{\theta} = \{\theta_1, \theta_2\}$, the probability density function of the Ising model is:

$$p(\mathbf{x}|\boldsymbol{\theta}) \propto \exp(-H(\mathbf{x}; \boldsymbol{\theta})),$$

$$H(\mathbf{x}; \boldsymbol{\theta}) = -\theta_1 \sum_{\langle i,j \rangle} x_i x_j - \theta_2 \sum_i x_i.$$

where $\langle i, j \rangle$ denotes that spin $i$ and spin $j$ are neighbours. $H$ is also called the Hamiltonian of the model. The likelihood function of this model is not analytical due to the intractable normalizing constant $Z(\boldsymbol{\theta}) = \sum_{\mathbf{x} \in \{-1,1\}^{m \cdot m}} \exp[-H(\mathbf{x}; \boldsymbol{\theta})]$. However, sampling from the model by MCMC is possible. Note that the sufficient statistics are known for this model: $s^*(\mathbf{x}) = \{\sum_{\langle i,j \rangle} x_i x_j, \sum_i x_i\}$. The true posterior can easily be approximated by rejection ABC with the low-dimensional sufficient statistics and extensive simulations. Here, we assume that $\theta_2$ is known, and the task is to infer the posterior of $\theta_1$ under an uniform prior $\theta_1 \sim \mathcal{U}(0, 1.5)$ (in this case the sufficient statistic becomes only 1D: $s^*(\mathbf{x}) = \sum_{\langle i,j \rangle} x_i x_j$). The true parameters are $\boldsymbol{\theta}^* = \{0.3, 0.1\}$.

In Figure 1(c), we investigate whether the proposed statistic could achieve sufficiency. Ideally, if the learned statistic $s$ in our method does recover the true sufficient statistic $s^*$ well, the relationship between $s$ and $s^*$ should be nearly monotonic (note that both $s$ and $s^*$ are here 1D). To verify this,

---

[3]We chose to conduct experiments on these models rather than common tasks like M/G/1 and Lotka-Volterra since they lack a known true likelihood, making it hard to verify the sufficiency of the proposed statistics. How to evaluate LFI methods on models without known likelihood is still an open problem (Lueckmann et al., 2021).

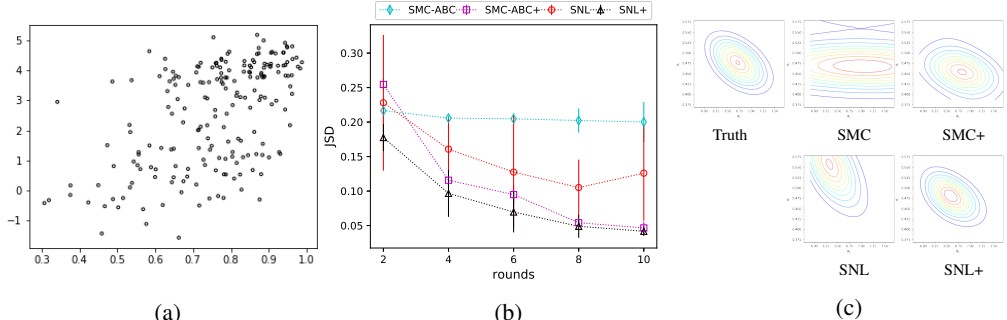

(a)            (b)            (c)

Figure 3: **Gaussian copula**. (a) The observed data $\mathbf{x}_o$ in this problem, which is comprised of a population of 200 i.i.d samples. (b) The JSD between the true/learned posteriors. (c) The contours.

| SMC' | SMC+ | SNL' | SNL+ | SRE | SNPE |
|---|---|---|---|---|---|
| $0.183 \pm 0.014$ | $0.047 \pm 0.009$ | $0.054 \pm 0.016$ | $0.042 \pm 0.006$ | $0.052 \pm 0.032$ | $0.037 \pm 0.018$ |

Table 2: **Gaussian copula**. The JSD between the learned and true posterior with 10,000 simulations. Here SMC' and SNL' utilize the hand-crafted summary statistics guided by human prior knowledge.

we plot the relationship between $s^*$ and $s$. We see from the figure that $s$ learned by our method does, approximately, increase monotonically with $s^*$, suggesting that $s$ recovers $s^*$ reasonably well. In comparison, the statistics learned with the widely-used posterior-mean-as-statistics approach only weakly depends on the true sufficient statistic; it is nearly indistinguishable for different $s^*$. In other words, it loses sufficiency. The result empirically verifies our theoretical result in Proposition 2.

Figure 1(b) further shows the JSD between the true and learned posterior for different methods across the rounds (the vertical lines indicates standard derivation, each JSD is obtained by calculating the average of 5 independent runs. The results shown in the below experiments have the same setup). It can be seen from the figure that for this model, likelihood-free inference methods augmented with the proposed statistic (SMC-ABC+, SNL+) outperform their original counterparts (SMC-ABC, SNL) by a large margin. In Table 1, we further compare our statistics with the expert designed statistics, from which one can see their close performance (here the expert statistics is taken as the true sufficient statistics $\mathbf{s}^*$). We further see that our method also outperforms SRE which directly estimates the ratio $t(\mathbf{x}, \boldsymbol{\theta}) = p(\mathbf{x}, \boldsymbol{\theta})/p(\mathbf{x})p(\boldsymbol{\theta}) \propto L(\boldsymbol{\theta}; \mathbf{x})$ in high-dimensional space (note that the true likelihood is of the form $L(\boldsymbol{\theta}; \mathbf{x}) = \exp(\boldsymbol{\theta}s^*(\mathbf{x}))/Z(\boldsymbol{\theta})$) as well as SNPE (version B). The reason why SNPE(-B) does not perform more satisfactorily might be that it relies on importance weights to facilitate sequential learning, which can induce high variance that makes the training unstable.

**Gaussian copula**. The second model we consider is a 2D Gaussian copula model (Chen & Gutmann, 2019). Data $\mathbf{x}$ for this model can be generated with aid of a latent variable $\mathbf{z}$ as follows:

$$\mathbf{z} \sim \mathcal{N}\left(\mathbf{z}; \mathbf{0}, \begin{bmatrix} 1, & \theta_3 \\ \theta_3, & 1 \end{bmatrix}\right),$$

$$x_1 = F_1^{-1}(\Phi(z_1); \theta_1), \quad x_2 = F_2^{-1}(\Phi(z_2); \theta_2),$$

$$f_1(x_1; \theta_1) = \text{Beta}(x_1; \theta_1, 2), \quad f_2(x_2; \theta_2) = \theta_2 \mathcal{N}(x_2; 1, 1) + (1 - \theta_2)\mathcal{N}(x_2; 4, 1/4).$$

where $\Phi(\cdot)$, $F_1(x_1; \theta_1)$, $F_2(x_2; \theta_2)$ are the cumulative distribution function (CDF) of the standard normal distribution, the CDF of $f_1(x_1; \theta_1)$ and the CDF of $f_2(x_2; \theta_2)$ respectively. We assume that a total number of 200 samples are i.i.d drawn from this model, yielding a population $\mathbf{X} = \{\mathbf{x}_i\}_{i=1}^{200}$ that serves as our observed data. Note that the likelihood of this model can be computed analytically by the law of variable transformation. To perform inference, we compute a rudimentary statistic to describe $\mathbf{X}$, namely (a) the 20-equally spaced quantiles of the marginal distributions of $\mathbf{X}$ and (b) the correlation between the latent variables $z_1, z_2$ in $\mathbf{X}$, resulting in a statistic of dimensionality 41. A uniform prior is used: $\theta_1 \sim \mathcal{U}(0.5, 12.5), \theta_2 \sim \mathcal{U}(0, 1), \theta_3 \sim \mathcal{U}(0.4, 0.8)$ and $\boldsymbol{\theta}^* = \{6, 0.5, 0.6\}$.

In Figure 2(b), we demonstrate the power of our neural sufficient statistic learning method on the Gaussian copula problem. Overall, we see that the proposed method improves the accuracy of

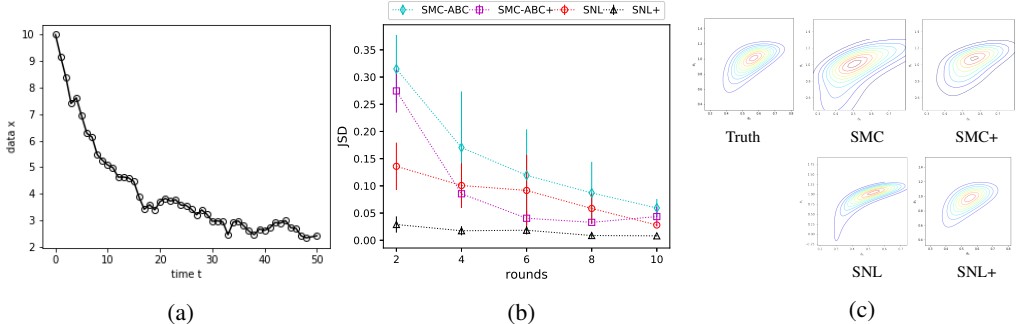

Figure 4: **OU process**. (a) The observed time-series data $\mathbf{x}_o = \{x_t\}_{t=1}^{50}$. (b) The JSD between the true and the learned posteriors. (c) The contours of the true posterior and the learned posteriors.

| SMC' | SMC+ | SNL' | SNL+ | SRE | SNPE |
|---|---|---|---|---|---|
| $0.040 \pm 0.006$ | $0.044 \pm 0.018$ | $0.004 \pm 0.001$ | $0.009 \pm 0.002$ | $0.022 \pm 0.013$ | $0.019 \pm 0.009$ |

Table 3: **OU process**. The JSD between the learned and the true posterior with 10,000 simulations. Here SMC' and SNL' utilize the hand-crafted summary statistics guided by human prior knowledge.

existing likelihood-free inference methods, as well as their robustness, see e.g. the reduced variability for SNL+ (the high variability in SNL may be due to the lack of training data required to learn the 41-dimensional likelihood function well). This is also confirmed by the contours plots in Figure 2(c). In Table 2 we further compare the proposed statistic with the expert-designed low-dimension statistic (here the expert statistic is taken to be the 5-equally spaced marginal quantiles and the correlations between $z_1, z_2$), from which we see that our proposed statistic achieves a better performance. For this model, the average performance of our method is slightly worse than that of SNPE. However, SNPE has a higher variability, so that the difference in performance is actually not significant.

**Ornstein-Uhlenbeck process**. The last model we consider is a stochastic differential equation (SDE). Data $\mathbf{x} = \{x_t\}_{t=1}^{D}$ in this model is sequentially generated as:

$$x_{t+1} = x_t + \Delta x_t,$$

$$\Delta x_t = \theta_1 (\exp(\theta_2) - x_t)\Delta t + 0.5\epsilon, \quad \epsilon \sim \mathcal{N}(\epsilon; 0, \Delta t).$$

where $D = 50$, $\Delta t = 0.2$ and $x_0 = 10$. This SDE can be simulated by the Euler-Maruyama method, and has an analytical likelihood. It has a wide application in financial mathematics and the physical sciences. Here, the parameters of interest are $\boldsymbol{\theta} = \{\theta_1, \theta_2\}$, and a uniform prior is placed on them: $\theta_1 \sim \mathcal{U}(0, 1), \theta_1 \sim \mathcal{U}(-2.0, 2.0)$. The true parameters are set to be $\boldsymbol{\theta}^* = \{0.5, 1.0\}$.

Figure 3(b) compares the JSD of each method against the simulation cost. Again, we find that the proposed neural sufficient statistics greatly improve the performance of both SMC-ABC and SNL. In Table 3, we compare our statistics to expert statistics (here the expert statistics are taken as the mean, standard error and autocorrelation with lag 1, 2, 3 of the time series). It can be seen that our statistics perform comparably to the expert statistics. Our method also seems to outperform SRE and SNPE.

## 6 CONCLUSION

We proposed a new deep learning-based approach for automatically constructing low-dimensional sufficient statistics for likelihood-free inference. The obtained neural approximate sufficient statistics can be applied to both traditional ABC-based and recent NDE-based methods. Our main hypothesis is that learning such sufficient statistics via the infomax principle might be easier than estimating the density itself. We verify this hypothesis by experiments on various tasks with graphs, i.i.d and sequence data. Our method establishes a link between representation learning and likelihood-free inference communities. For future works, we can consider further infomax representation learning approaches, as well as more principle ways to determine the dimensionality of the sufficient statistics.

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

# A  THEORETICAL PROOFS

## A.1  PROOF OF PROPOSITION 1

*Proof.* Firstly, assume $s(\cdot)$ is a sufficient statistic. By the definition of sufficient statistic we know $p(\mathbf{x}|\boldsymbol{\theta}) = p(\mathbf{x}|\mathbf{s})p(\mathbf{s}|\boldsymbol{\theta})$. Then we have the Markov chain $\boldsymbol{\theta} \to \mathbf{s} \to \mathbf{x}$ for the data generating process. On the other hand, since $\mathbf{x} \sim p(\mathbf{x}|\boldsymbol{\theta})$ and $S$ is a deterministic function we have the Markov chain $\boldsymbol{\theta} \to \mathbf{x} \to \mathbf{s}$. By data processing inequality we have $I(\boldsymbol{\theta}; s(\mathbf{x})) \leq I(\boldsymbol{\theta}; \mathbf{x})$ for the first chain and $I(\boldsymbol{\theta}; \mathbf{x}) \leq I(\boldsymbol{\theta}; s(\mathbf{x}))$ for the second chain. This implies that $I(\boldsymbol{\theta}; \mathbf{x}) = I(\boldsymbol{\theta}; s(\mathbf{x}))$ i.e $s$ is the maximizer of $I(\boldsymbol{\theta}; S(\mathbf{x}))$. For the other direction, since $I(\boldsymbol{\theta}; s(\mathbf{x})) = \max_S \ I(\boldsymbol{\theta}; S(\mathbf{x}))$, we have $I(\boldsymbol{\theta}; s(\mathbf{x})) = I(\boldsymbol{\theta}; \mathbf{x})$. Note that $\boldsymbol{\theta} \to \mathbf{x} \to \mathbf{s}$ is a Markov chain, from Theorem 2.8.1 of Cover et al. (2003) we can get $\boldsymbol{\theta}$ and $X$ is conditionally independent given $\mathbf{s}$. This implies $s$ is sufficient. ☐

## A.2  PROOF OF PROPOSITION 2

*Proof.* We can write the objective as $\mathbb{E}_{p(\boldsymbol{\theta},\mathbf{x})}[\|S(\mathbf{x}) - \boldsymbol{\theta}\|_2^2] = \int p(\boldsymbol{\theta}, \mathbf{x}) \log e^{\|S(\mathbf{x})-\boldsymbol{\theta}\|_2^2} d\mathbf{x} d\boldsymbol{\theta}$. On the other hand we have $I(\boldsymbol{\theta}; S(\mathbf{x})) = \int p(\boldsymbol{\theta}, \mathbf{x}) \log p(S(\mathbf{x})|\boldsymbol{\theta})/p(S(\mathbf{x})) d\mathbf{x} d\boldsymbol{\theta}$. By comparing them, we see they are generally not equivalent. Equivalence only holds in special cases (e.g. Gaussians). ☐

# B  MORE EXPERIMENTAL DETAILS AND RESULTS

## B.1  DETAILED EXPERIMENTAL SETTINGS

**Neural networks settings**. For the statistic network $S$ in our method (for both JSD and DC estimators), we adopt a $D$-100-100-$d$ fully-connected architecture with $D$ being the dimensionality of input data and $d$ the dimensionality of the statistic. For the network $H$ used to extract the representation of $\boldsymbol{\theta}$, we adopt a $K$-100-100-$K$ fully-connected architecture with $K$ being the dimensionality of the model parameters $\boldsymbol{\theta}$. For the critic network, we adopt a $(d + K)$-100-1 fully connected architecture. ReLU is adopted as the non-linearity in all networks. For SRE, which is closely related to our method, we use a $(D + K)$-144-144-100-1 architecture. This architecture has a similar complexity as our networks. All these neural networks are trained with Adam (Kingma & Ba, 2014) with a learning rate of $1 \times 10^{-4}$ and a batch size of 200. No weight decay is applied. We take 20% of the data for validation, and stop training if the validation error does not improve after 100 epochs. We take the snapshot with the best validation error as the final result.

For the neural density estimator in SNL/SNPE, which is realized by a Masked Autoregressive Flow (MAF) (Papamakarios et al., 2017), we adopt 5 autoregressive layers, each of which has two hidden layers with 50 tanh units. This is the same settings as in SNL. The MAF is trained with Adam with a learning rate of $5 \times 10^{-4}$ and a batch size of 500 and a slight weight decay ($1 \times 10^{-4}$). Similar to the case of MI networks, we take 20% of the data for validation, and stop training if the validation error does not improve after 100 epochs. The snapshot with the best validation error is taken as the result.

**Sampling from the approximate posterior/learnt proposal**. For fair comparison, we adopt simple rejection sampling for all LFI methods (ABC, SNL, SNPE, SRE) when sampling from the learnt posterior, so that each LFI method only differs in the way they learn the posterior. No MCMC is used.

**Empirical version of objective functions**. Recall that in the JSD estimator, the statistic network $S(\cdot)$ is trained with the following objective together with the critic network $T(\cdot)$:

$$\text{maximize}_{S,T} \ \mathcal{L}(S, T) = \mathbb{E}_{p(\boldsymbol{\theta},\mathbf{x})} \left[ -\text{sp}\left(-T(\boldsymbol{\theta}, S(\mathbf{x}))\right) \right] - \mathbb{E}_{p(\boldsymbol{\theta})p(\mathbf{x})} \left[ \text{sp}\left(T(\boldsymbol{\theta}, S(\mathbf{x}))\right) \right]$$

the mini-batch approximation to this objective is:

$$\mathcal{L}(S, T) \approx \frac{1}{n} \sum_i^n \left[ -\text{sp}\left(-T(\boldsymbol{\theta}_i, S(\mathbf{x}_i))\right) \right] - \frac{1}{m} \frac{1}{n} \sum_j^m \sum_i^n \left[ \text{sp}\left(T(\boldsymbol{\theta}_{j_i}, S(\mathbf{x}_i))\right) \right]$$

where $\{j_1, j_2, ..., j_n\}$ is the $j$-th random permutation of the indexes $\{1, 2, ..., n\}$ and the pair $(\boldsymbol{\theta}_i, \mathbf{x}_i)$ are randomly picked from the data $\mathcal{D} = \{\boldsymbol{\theta}_i, \mathbf{x}_i\}_{i=1}^N$. Here we set $m = 400$ and $n$ is the batch size.

In the DC estimator, the statistic network is trained by the following objective:

$$\text{maximize}_S \ \mathcal{L}(S) = \frac{\mathbb{E}_{p(\boldsymbol{\theta},\mathbf{x})p(\boldsymbol{\theta}',\mathbf{x}')}[h(\boldsymbol{\theta}, \boldsymbol{\theta}')h(S(\mathbf{x}), S(\mathbf{x}'))]}{\sqrt{\mathbb{E}_{p(\boldsymbol{\theta})p(\boldsymbol{\theta}')}[h^2(\boldsymbol{\theta}, \boldsymbol{\theta}')]} \cdot \sqrt{\mathbb{E}_{p(\mathbf{x})p(\mathbf{x}')}[h^2(S(\mathbf{x}), S(\mathbf{x}'))]}},$$

where $h(\mathbf{a}, \mathbf{b}) = \|\mathbf{a} - \mathbf{b}\| - \mathbb{E}_{p(\mathbf{b}')}[\|\mathbf{a} - \mathbf{b}'\|] - \mathbb{E}_{p(\mathbf{a}')}[\|\mathbf{a}' - \mathbf{b}\|] + \mathbb{E}_{p(\mathbf{a}')p(\mathbf{b}')}[\|\mathbf{a}' - \mathbf{b}'\|]$. The mini-batch approximation to this objective is:

$$\mathcal{L}(S) \approx \frac{\sum_{i,j}^{n,n} \tilde{h}(\boldsymbol{\theta}_i, \boldsymbol{\theta}_j)\tilde{h}(S(\mathbf{x}_i), S(\mathbf{x}_j))}{\sqrt{\sum_{i,j}^{n,n} \tilde{h}^2(\boldsymbol{\theta}_i, \boldsymbol{\theta}_j)} \cdot \sqrt{\sum_{i,j}^{n,n} \tilde{h}^2(S(\mathbf{x}_i), S(\mathbf{x}_j))}},$$

where $\tilde{h}(\mathbf{a}_i, \mathbf{b}_j) = \|\mathbf{a}_i - \mathbf{b}_j\| - \frac{1}{n-2}\sum_{j'}^{n} \|\mathbf{a}_i - \mathbf{b}_{j'}\| - \frac{1}{n-2}\sum_{i'}^{n} \|\mathbf{a}_{i'} - \mathbf{b}_j\| + \frac{1}{(n-1)(n-2)} \sum_{i',j'}^{n,n} \|\mathbf{a}_{i'} - \mathbf{b}_{j'}\|$. Here $i, j, i', j'$ are the indexes in the mini-batch. $n$ is again the batch size.

**JSD calculation between true posterior and approximate posterior**. The calculation of the Jensen-Shannon divergence between the true posterior $P$ and approximate posterior $Q$, namely $\mathrm{JSD}(P, Q) = \frac{1}{2}\mathrm{KL}[P\|(P+Q)/2] + \frac{1}{2}\mathrm{KL}[Q\|(P+Q)/2]$, is done numerically by a Riemann sum over $30^K$ equally spaced grid points with $K$ being the dimensionality of $\boldsymbol{\theta}$. The region of these grid points is defined by the min and max values of 500 samples drawn from $P$. When we only have samples from the true posterior (e.g. the Ising model), we approximate $P$ by a mixture of Gaussian with 8 components.

### B.2 ADDITIONAL EXPERIMENTAL RESULTS

**Comparison of different MI estimators**. We compare the performances of four MI estimator for infomax statistics learning: Donsker-Varadhan (DV) estimator (Belghazi et al., 2018), Jensen-Shannon divergence (JSD) estimator (Hjelm et al., 2018), distance correlation (DC) Székely et al. (2014) and Wasserstein distance (WD) (Ozair et al., 2019). We highlight that the last two estimators (DC and WD) are ratio-free. We compare the discrepancy between the true posterior and the posterior inferred with the statistics learned by each estimator, as well as the execution time per each mini-batch. The results, which are averaged over 5 independent runs, are shown in the figure and the table below.

From the figure we see that the JSD estimator generally yields the best accuracy among the four estimators. In terms of execution time, the DC estimator is clearly the winner, with its execution time being only 1/15 of the other estimators. However, the accuracy of the DC estimator is still comparable to the JSD estimator, especially when the number of training samples is large (e.g. 10,000). According to these results, we suggest using JSD in small-scale settings (e.g. early rounds in sequential learning), and use DC in large-scale ones (e.g. later rounds in sequential learning).

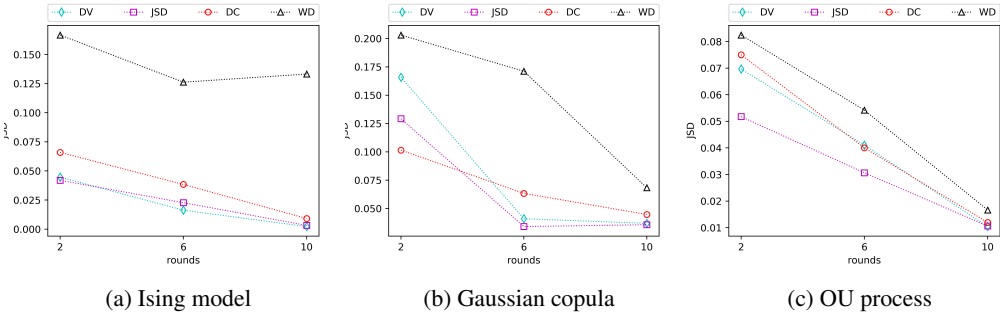

| (a) Ising model | (b) Gaussian copula | (c) OU process |

Figure 5: Comparing the accuracy of different MI estimator for infomax statistics learning.

| Ising model | | | | Gaussian copula | | | | OU process | | | |
|---|---|---|---|---|---|---|---|---|---|---|---|
| DV | JSD | DC | WD | DV | JSD | DC | WD | DV | JSD | DC | WD |
| 115 | 124 | 6 | 230 | 154 | 167 | 10 | 288 | 143 | 158 | 13 | 256 |

Table 4: Comparing the execution time (ms) of different MI estimator for infomax statistics learning.

**Contrastive learning v.s. MLE**. In the experiment in the main text, we discover that our method does not always achieve the best performance; it does not work better than SNPE-B on the Gaussian copula problem. Here we would like to investigate why this happens.

Upon a closer look, we discover that SRE, which is closely related to our method when used with the JSD estimator, is outperformed by SNPE-B on the Gaussian copula problem. Remark that both SRE and our method, when used with the JSD estimator, uses contrastive learning rather than MLE. Since both of these two contrastive learning methods do not perform better than the MLE-based SNPE-B, it makes us suspect the reason is due to imperfect contrastive learning. To verify this, we further conduct experiments for SNPE-C, which shares the same loss function with SRE but with a different parameterization to the density ratio (SRE: fully-connected network; SNPE-C: NDE-based parameterization. This NDE is the same as in SNL). The result is as follows:

| Ising model | | | | Gaussian copula | | | | OU process | | | |
|---|---|---|---|---|---|---|---|---|---|---|---|
| SRE | SNPE-B | SNPE-C | SNL+ | SRE | SNPE-B | SNPE-C | SNL+ | SRE | SNPE-B | SNPE-C | SNL+ |
| 0.083 | 0.058 | 0.030 | 0.017 | 0.052 | 0.037 | 0.047 | 0.042 | 0.022 | 0.018 | 0.016 | 0.009 |

Table 5: Commparing the the JSD of contrastive learning-based methods (SRE, SNPE-C, SNL+) and MLE-based method (SNPE-B) on the three models considered in the experiments in the main text.

Surprisingly, we find that SNPE-C also perform less satisfactorily than SNPE-B on the Gaussian copula problem. This suggests that contrastive learning might be less preferable than MLE on the Gaussian copula problem, which might also explain the less satisfactory performance of our method.

