# OpenReview forum: "Neural Approximate Sufficient Statistics for Implicit Models"
_ICLR.cc/2021/Conference — ICLR 2021 Spotlight_

### Official Review · AnonReviewer3 · 2020-10-23
**A promising ABC method with selection of statistics optimizing an information-theoretic criterion**

**Rating:** 7
**Confidence:** 4

**Review:**

The paper considers the problem of coming up with summary statistics for ABC-type approximations.  The strategy is to find a function of the data S(X) parametrized by neural networks, with the objective of maximizing a mutual information criterion.  The motivation comes from an interesting classical result described by Shamir, Sabato and Tishby, relating sufficiency and mutual information, combined with a machinery inspired by Hjelm et al, "Learning  deep representations  by  mutual  information estimation and  maximization" published in ICLR 2019, and Monte Carlo algorithms to learn the ABC posterior and the statistics jointly. The method is illustrated on toy examples.

The topic is very interesting and the paper contains a real contribution. It is particularly remarkable as a lot of articles have been written on the topic of the summary statistics for ABC, without much progress in the last few years.  The use of information-theoretic consideration is fruitful to define a meaningful concept of "near sufficiency". Indeed we know that exact sufficiency is only attainable in exponential models, while ABC methods are intended for much more complex settings. The experiments are only dealing with simple models, not really typical of challenging settings for which ABC methods are useful, but they make a convincing case that the method is promising.

It might be worth reminding the readers of the Pitman-Koopman-Darmois theorem on sufficient statistics of fixed dimension in the exponential family, and perhaps a few more elements from Shamir et al 2008 could be recalled to motivate the relaxed notion of "near sufficiency".  There could also be more comments on the use of JSD instead of KL, as in Appendix A of Hjelm et al, as this seems important.  I would find these reminders more useful and informative than the authors' remarks on the curse of dimensionality, which seem less central to the manuscript. The authors mention a research direction around eq (7) related to the dimension of the statistic but leave it for future work.

Proposition 2 itself is not very surprising, given that the "posterior mean as statistic" is not designed to minimize the same information-theoretic loss, and given the lack of fixed-dimensional sufficient stats in generic models. On the other hand the posterior mean is optimal for another loss, as shown in Theorem 3 of Fearnhead & Prangle. I find the writing around this proposition to be a bit partial; it would be preferable to use neutral terms and to explain that the different approaches optimize different losses.

The SMC-ABC looks more like a population Monte Carlo algorithm as in Beaumont et al., or an adaptive importance sampling strategy than an SMC sampler. In SMC the particles are typically propagated across steps, weighted and resampled, whereas here, at each iteration of Algorithm 1, particles on the parameter space are drawn from a parametric distribution $p_j$; they are not propagated from the previous $\theta$'s.

I was not sure why Figure 1c was looking so wiggly?

The exact posterior in the Ising model could be obtained from the exchange algorithm or similar techniques for distributions with intractable normalising constants, instead of the suggested rejection sampler. See e.g.
Murray, Ghahramani & MacKay, MCMC for doubly-intractable distributions.
Moeller, Pettitt, Berthelsen & Reeves, An efficient Markov chain Monte Carlo method for distributions with intractable normalising constants.

The discretization of the OU process is also known as an autoregressive process of order 1, although the parametrization is typically different. I wonder if the choice of 'expert' statistics could be commented on further. Are the mean, sd and autocorrelation of order 1 actually sufficient here?

On the last page, "For future works, we can consider other infomax approaches." sounds a bit shallow.

In the experiments, instead of selecting a subset of the order statistics, I wonder if the authors have tried to use all the order statistics. The distance between order statistics actually might be a very sensible way of comparing marginal distributions... the "reduction of dimension" associated with the selection of some order stats might not actually bring much gains.

Finally, how does the method relate to theoretical works on ABC, such as Frazier et al or Li & Fearnhead? Do the automatically derived summary statistics satisfy the assumptions of these papers? If so, something could perhaps be said about asymptotic concentration of the resulting posteriors?

---

> ### Author Response · Authors · 2020-11-21
> **Thanks for your insightful and inspiring comments**
>
> Thank you for the fruitful comments. They really help to improve our work. We have revised the paper accordingly:
>
>
> - PKD theorem and justifying the use of JSD: we thank the reviewer for their valuable suggestions and have revised the manuscript accordingly. The PKD theorem also better motivates our eq 7 in our manuscript that one shall determine the dimensionality of the statistics in a data-driven way, and we will add a discussion about this in the conclusion/future works part.
>
>
> - Proposition 2: it was not our intention to add a spin to our description. Thank you for pointing that it could be read that way; the advice that objectively viewing different approaches as different targets is indeed very wise. In fact, our method targets more at the full posterior whereas the mean-as-statistics approach targets more at point estimate. We will revise the paper accordingly.
>
>
>
> - SMC-ABC: it is true that the SMC-ABC algorithm described in the manuscript does seem to be closer to PMC-ABC. We used the umbrella term 'SMC-ABC' to describe a family of ABC algorithms that sequentially learn the posterior rather than a specific algorithm, which include both SMC-ABC and PMC-ABC. In fact the boundary between SMC/PMC-ABC is blurry in many papers. Please let us know if you still feel this naming inappropriate :)
>
>
> - Regarding the wiggly Figure 1: we think this is because the proposed infomax statistics, while effective, is still not perfect that it does not exactly recover the true sufficient statistics. This can also be seen from Table 1 when compared to the expert (true) sufficient statistics. This reflects that there is still room for improvement in our method.
>
>
> - Exact posterior in Ising model: we greatly appreciate the reviewer for pointing us to the two MCMC works for sampling from distributions with intractable normalizing constants, which should definitely be more effective and should be mentioned.
>
>
>
> - The statistics for discretized OU process: thanks a lot for reminding us of its equivalence to the AR(1) model. We have checked that the expert statistics used here, which is also widely used in time-series analysis, is actually not the sufficient statistics for AR(1), although it  achieves very good inference quality. Please also note that our method can select between expert/learnt statistics (by the MI criteria) and/or to even combine them (by feeding them jointly to the $T$ network), so even if our learnt statistics performs slightly worse than expert statistics it would not be an problem.
>
>
> - Regarding the order statistics: we guess you are talking about the Gaussian copula problem. Actually in the experiment there we have used nearly all the order statistics (the 20 equally spaced quantiles of the marginal distribution) exactly as you suggested. Or have we just simply missed something?
>
> - Asymptotic results: thank you very much for pointing us to these theoretical works! After a check we think that some of the condition there might not be satisfied in our work. For example, the asymptotic results in Li \& Fearnhead's rely on CLT which itself relies on condition 4 in their paper, which requires the observed data to be i.i.d. This might not be true in cases with graph or sequence data (e.g Ising model and OU process).

---

### Official Review · AnonReviewer2 · 2020-10-27
**Nice paper but would benefit from a thorougher discussion of similar work.**

**Rating:** 7
**Confidence:** 3

**Review:**

# Summary
The paper presents a method for building sufficient statistic $\mathbf{s}(\mathbf{x})$ of observed data $\mathbf{x}$ for a parameter $\theta$. It focuses on using this statistic for performing posterior inference in a likelihood-free setting. In particular, the authors proposed to improve SMC-ABC and SNL with their statistics. The results suggest the method is able to learn good statistics and compares favorably with other LFI algorithms.
# Major comments
## Pros:
The paper is well written and easy to follow. Moreover, the introduction of SMC-ABC and SNL+ is interesting as it improves the performance of the classic algorithms. Although the training procedure in itself is not new I think (see comments below), finding a sufficient statistic is conceptually new and is very relevant.
## Cons:
1) Formally, your method is very close to SRE. Indeed the MI loss used to learn the statistic is equivalent to the binary cross-entropy used by https://arxiv.org/pdf/1903.04057.pdf [1] to learn the discriminator. In the end, the network T can be seen as a discriminator (without the sigmoid activation in the computation of the loss) between (x, \theta) pairs that only takes as input s(x), a compressed version of x. Thus the novelty is not really about how you learn the sufficient statistic but more the way you use it. In [1], they use the ratio whereas you directly plug the statistic into SMC-ABC and SNL.
Thus when discussing MI and RE just before the experiments section, I do not agree with your arguments 1) and 2) are just an architectural choice that can be made.
2) I wonder whether discussing Bayesian Sufficient statistics couldn't be interesting?
3) You don't write about minimal sufficient statistics whereas this is exactly what you are looking for as it is the one that will provide the best SNR.
4) Why don't you compare to more standard tasks such as the one used by SNL or SRE?
5) I did not get the point about monotonicity between two sufficient 1D statistics, could you explain why this should be correct? Indeed
"A statistic T = T (x) is said to be minimally sufficient for the parameter $\theta$ if it is sufficient for $\theta$ and for any other sufficient statistic S = S (x) there exists a function g(.) with T(x) = g(S(x))." Nothing prevents g to be something else than monotonic. The only case where this is maybe true is if S and T are minimal.

## What could be made to address my comments

1) You should clarify the similarities between SRE and your work, showing they share the same training procedure but that you are more specific about the architectural choice. I agree that your work is original and conceptually distinct from SRE. However, the only formal difference between the two is that you specify how to input the parameters and data into the neural network whereas SRE does not. The two algorithms introduced in this paper are original and seem very competitive and thus the novelty seems to be there more than anywhere else.
2) This is just a question!
3) I think the "minimal sufficient statistic" should be defined somewhere, also discussing the impact of just finding a sufficient statistic in contrast to a minimal one should be discussed. Probably, ABC and SNL are able to discard the "useless" noise from the statistic but I suppose this has somewhere an impact on the sample efficiency.
4) Compare on SLCP, Lotka-Volterra, etc...
5) Just clarify this point.

*I will increase my score to 7 or 8 depending on how you address my comments (or convince me they don't make sense).*

## Minor comments
Figure 1: (b) What about SRE? (c) could you improve the quality of this plot, maybe creating it as a pdf?
page 6: "Note that the sufficient statistic" -> "a sufficient"?

---

> ### Author Response · Authors · 2020-11-21
> **Thank you for your positive remarks and detailed inquiries; our method is more than an architecture choice**
>
>
> 1: Regarding the arguments 1) and 2) discussing MI and RE:
>
> - For argument 1), it is true that our current method is related to an architecture choice in ratio estimation methods (split architecture v.s fully-connected). However, we would like to stress that our paper is more about learning summary statistics based form principles of information theory. Mutual information is connected to density ratio estimation, and hence the connection to the RE methods, as you point out. That said, the architecture choice is indeed an important inductive bias that the data $x$ and the model parameter $\theta$ should not interact with each other directly but their representations, based on a sufficient statistics prospective. This prospective was overlooked in existing LFI ratio literature. Such a perspective also provides us with a way to readily compare different statistics (e.g expert v.s learnt, by using the proposed infomax criteria) or to even combine them (e.g expert $\cup$ learnt, by feeding them jointly to the $T$ network in our method). This is also very different to existing ratio estimators.
>
> - For argument 2), we think this cannot be simply reduced to an architecture choice. This is because there do exist some infomax techniques that never return the ratio (e.g Wasserstein-based one [1]  or score-based one [2]), and when using these techniques to learn the statistics we will no more estimate any ratio, not to speak of an architecture choice in ratio estimators.
>
> $\quad$
>
> 2: Surely! In fact eq.(2) in the manuscript has (implicitly) used the notation of Bayesian sufficient statistics. We are glad that we share this point with the reviewer :)
>
> $\quad$
>
> 3: The suggestion on discussing minimal sufficient statistics is very insightful. But what we are seeking is more like 'near-sufficient statistics with the lowest dimensionality', which is slightly different from minimal sufficient statistics (remark: two minimal sufficient statistics $s_1$, $s_2$ could have different dimensionalities, for example $s_1=(x, y)$ and $s_2=(x, y, xy)$). We will provide a discussion/clarification about this in the revised manuscript.
>
> $\quad$
>
> 4: We also agree that the tasks used in SNL and SRE (e.g M/G/1 and Lotka-Volterra) are useful but might not be so appropriate in our case. This is because these tasks do not have known posterior, and the current evaluation metric $\log p(\theta^*|x_o)$ (here $\theta^*$ is the true parameter and $x_o$ is the observed data) used in SNL and SRE on these tasks are solely a point estimate. In such situations, it seems impossible to establish whether the proposed statistic is sufficient or not.
>
> $\quad$
>
> 5: It turns out that if two 1D statistics $s_1$, $_2$ are both (near-)sufficient then they should be both minimal sufficient, which then coincides with your insight (they are monotonic w.r.t each other)?
>
> $\quad$
>
> Minor comments: thanks for the suggestions, we shall improve our writing in our revision.
>
> $\quad$
>
> [1] Ozair et.al. Wasserstein dependency measure for representation learning, NeurIPS 19.
>
> [2] Wen et. al Mutual information gradient estimation for representation learning, ICLR 20.

---

> > ### Comment · AnonReviewer2 · 2020-11-23
> > **Thanks for clarifications.**
> >
> > 1. Ok I agree that you are seeking for something that could be different depending on the kind of metric used. However, in practice, you have used eq 4 which is equal to the loss used in RE. I really believe making the parallel with RE explicit would help the reader to better understand where lies your contribution and how it differs from other works. The discussion about combining it with domain knowledge-based statistics is interesting as well.
> > 2. Similar to point 3 I think you could make that clearer then. :)
> > 3. Yes I think this is something that should be clarified!
> > 4. Ok it seems legit.
> > 5. Thanks for the clarification, I now understand! :)
> >
> > Overall authors addressed my comments. I would still argue for further discussions about RE and sufficient statistics than the one present in the paper right now.
> >
> > As promised I will increase my score to 7.

---

> > > ### Author Response · Authors · 2020-11-23
> > > **Thanks a lot for your feedback as well as increasing the score**
> > >
> > > 1: We will discuss explicitly the similarity between our method and RE in the revised paper, which as you said should help to better understand our true contribution. In fact your viewpoint on architecture choice is very concise and accurate; we will use it in the revised paper. After discussing the similarity to RE we would then discuss their difference, including (a) how our method provides a additional sufficient statistics-based prospective and (b) the non-ratio estimating case, as we discussed above.
> > >
> > > 2-5: We will make things clear and explicit as you suggested :)
> > >
> > > Thanks very much for your fruitful feedback and advice again!

---

### Official Review · AnonReviewer1 · 2020-10-28
**Potentially impactful, weaknesses in empirical comparison, conceptual question**

**Rating:** 6
**Confidence:** 5

**Review:**

### Summary
The paper introduces a new method for learning approximately sufficient summary statistics in the context of likelihood-free inference. To do so, neural networks are trained with a loss maximising mutual information, using a JSD surrogate estimator previously proposed in the literature. The authors show how their approach can for example be combined with SMC-ABC or SNL, leading to marked performance improvements on three problems.

### Score
I appreciate the core idea of the paper as well as the empirical improvements over SMC-ABC and SNL. As detailed below, my main concern is the comparison to existing LFI approaches that are able to automatically reduce data dimensionality (SRE, SNPE). I hope the authors will address the weaknesses and conceptual question during rebuttal. As it stands, I view the paper as marginally below acceptance.

### Strengths
+ The proposed method of learning summaries is novel and very relevant to many applications of LFI
+ SMC-ABC+ and SNL+ show marked performance improvements
+ Comparison of different mutual information estimators
+ Concise discussion of their approach versus posterior-as-mean statistic used in some previous work
+ Comparison to hand-crafted summary statistics (SMC-ABC', SNL')

### Weaknesses
As mentioned above, I see the main weakness in the comparison to SRE and SNPE, which falls short in several ways:
- The authors include comparison to SNPE-B rather than SNPE-C arguing that “We select to compare with SNPE-B (Lueckmann et al., 2017) rather than the more recent SNPE-C (Greenberg et al., 2019) due to its equivalence to SRE shown in (Durkan et al., 2020)”. While Durkan et al. 2020 indeed show an equivalence in the loss functions of both approaches, this does not mean that they are the same and comparison is not warranted: While SNPE-C/APT trains a density estimator to estimate the posterior directly, a classifier architecture is used for SRE/AALR, combined with MCMC sampling. While these two approaches share similarity in the loss, they perform differently in practice, due to different training problems/inductive biases/algorithmic steps. Therefore, results using the newer SNPE-C version should be reported (which generally outperforms SNPE-B)
- No details on how SRE and SNPE-B were used are reported in the paper, i.e., it is unclear which network architectures were used, and whether these were appropriate for the problems at hand. I briefly checked the code in the supplementary material, but could not find hyperparameters for these methods in the respective notebooks either. In any case, this is central information which needs to be reported properly in the papers’ appendix
- OU process: Judged by MMD instead of JSD, the performance of SRE and SNPE seems comparable to SNL+ on the OU model (Figure 6 Appendix), while this is not the case when looking at JSD results in Table 3. How can the discrepancy be explained?

More generally, regarding empirical comparisons:
- In the figures reporting results it is not defined what error bars represent
- Have the authors checked whether JSD calculation is stable, i.e., yields similar results, when using a finer grid and more than 500 samples?
- Results on the MMD metric for the Ising model are missing from the appendix
- It would have been good to include a truly high-d problem, e.g. involving images

### Conceptual question

In the appendix, the authors show the posterior directly extracted from the critic. How exactly were the plots shown for comparison and the JSDs calculated, and more specifically, could the difference in JSD be explained by calculating the normaliser on coarse grid? As a control, I would like to see a sample-based comparison, i.e., a comparison of samples from the approximate posterior extracted from T (e.g., by MCMC) versus samples generated from the NDE refit. If it turns out that results are comparable, wouldn't it make sense to think of the algorithm as a new LFI algorithm in itself rather than a pre-processing step reducing dimensionality of data for other algorithms?


### Additional comments

- As a suggestion: It might make sense to first compare and discuss SMC, SMC+, SMC’ as well as  SNL, SNL+, SNL’, and second, go on to compare SMC+, SNL+ versus SRE, SNPE


### Minor comments

- Figure 5 and 6 (Appendix): SNP should be SNPE for consistency with main paper
- Figure 6 (Appendix): OP should be OU for consistency with Figure 3 (main paper)
- Ticks in Figures 2c and 3c are too small to be legible
- The reference section should be checked carefully, to name a few examples: Hermans et al. (2019) has been published, Durkan et al. (2020) has been published, Kingma et al. (2014) has been published, Aaron Van Den Oord et al. (2016) misses a venue, Aaron van den Oord (2017) has been published, Rippel et al. (2014) misses a venue, Thomas et al. (2016) has been published


### Update

The authors have addressed most of the above criticisms. Even though part of the definite answer about directly extracting the posterior from the MI network is left to future work and no test of their method on a high-d example was included, I believe that the paper is a valuable addition to the literature. I have increased my score accordingly to acknowledge the authors' replies.

---

> ### Author Response · Authors · 2020-11-21
> **Thanks for your valuable criticism; we have provided results for SNPE-C and also more experimental details**
>
>
>
> Thank you for these detailed comments and questions. They identify some important points which we hope to clarify and address here and in our revision.
>
> Regarding the weakness:
>
> - SNPE-C v.s SRE: Thank you for pointing out the differences between them! We should definitely also compare to SNPE-C. We initially choose to compare to SNPE-B rather than C because it is not based on contrastive learning, and hence increases the diversity of the baselines. Now we have provided below the results for SNPE-C. Here we adopt the (re-formulated) algorithm described in Durkan et. al, which shares the same loss function with SRE but with a different parameterization to the density ratio (SRE: fully-connected network; SNPE: NDE-based parameterization. This NDE is exactly the same as in SNL). Surprisingly, we found that although SNPE-C does improve over SNPE-B on Ising model and OU process, it is less satisfactory on the Gaussian copula problem. We also see the three contrastive learning-based methods (SRE, SNPE-C, SNL+) all perform less satisfactorily than SNPE-B. This might also somewhat explain why our method does not outperform SNPE-B on this problem. According to the results, we feel it is better for us to keep reporting the SNPE-B results in the manuscript, and we will add the results for SNPE-C to the appendix.
>
> .................................................................
>
> Ising  / Gaussian copula  / OU process
>
> SRE:        0.083 / 0.052 / 0.022
>
> SNPE-B: 0.058 / 0.037 / 0.018
>
> SNPE-C: 0.030 / 0.047 / 0.016
>
> SNL+    : 0.017 / 0.042 /  0.009
>
> .................................................................
>
>
>
>
> - Details on SRE and SNPE: we have now reported them in the appendix. More specifically, for SRE, we use the same optimization settings and similar number of network parameters as in our MI-based statistics learning network. For SNPE we use exactly the same NDE/optimization settings as in SNL.
>
> - Problem with MMD: we agree that the calculation of MMD might be less reliable. This is also raised by other reviewers. Please also check our response "issues with empirical evaluation" to R4.
>
> - General problems in empirical evaluation: thanks so much for pointing out them. We will revise the paper accordingly.
>
> Regarding the conceptual question:
>
> Your question is very insightful and is indeed very important. Let us discuss it from two aspects:
>
> - We completely agree that the inaccurate calculation of the normaliser might affect the result. To avoid this, we have re-checked the result using a finer grid, and the conclusion seems to be the same. We conjecture the underlying reason might be that 'being correctly normalized by construction' (NDE-refit) and 'post-adjusting the learnt density so that it is normalized' (sampling from the extracted ratio by e.g MCMC) is different, as the former has a helpful inductive bias during learning, which yields better performance;
>
> - We also agree that one can view the infomax phase itself as a LFI algorithm (an improved ratio estimator). However, by viewing the infomax phase as sufficient statistics learning rather than density estimate, we have the freedom to use any infomax methods to learn the statistics, even if it is ratio-free (e.g [1] and [2]), and the learnt statistics can be used in a wide range of LFI algorithms. This sufficient statistics viewpoint also points us to a way to readily compare/select different statistics (e.g expert v.s learnt, by the proposed infomax criteria). This is very different from existing density estimators.
>
> Other comments:
> Thanks a lot for pointing out them. We would address them in the revision.
>
> [1] Ozair et.al. Wasserstein dependency measure for representation learning, NIPS 19.
>
> [2] Wen et. al Mutual information gradient estimation for representation learning, ICLR 20.

---

> > ### Comment · AnonReviewer1 · 2020-11-24
> > **Stability of JSD calculation**
> >
> > Thank you for the reply, which addresses most of my concerns. I would highly appreciate if you could briefly address the following point which was left unanswered as well:  Have the authors checked whether JSD calculation is stable, i.e., yields similar results, when using a finer grid and more than 500 samples?

---

> > > ### Author Response · Authors · 2020-11-24
> > > **On the Stability of JSD calculation**
> > >
> > >
> > > Sorry for missing this point. Yes we can confirm that the JSD calculation is stable. The $30^K$ grids used here ($K$ is the number of model parameters/dimensionality of the posterior) seems to be enough for the three models considered in the experiment (where K=1,3,2 respectively); using a finer grid (e.g $50^K$ grids) indeed yields a simialr result.
> > >
> > > We are a bit not sure about what you mean by 'and more than 500 samples' in your question, since the JSD is directly calculated from two analytical densities p and q (here p = true posterior, q = posterior approximated by e.g NDE/ratio estimator). No samples are needed to be drawn from p and q in order to calculate the JSD.  Or have we just simply missed something?
> > >
> > > Again thanks very much for your valuable comments :)

---

> > > > ### Comment · AnonReviewer1 · 2020-11-24
> > > > **Thanks!**
> > > >
> > > > OK, great, thank you for the swift reply!
> > > >
> > > > I was referring to this statement in the description of JSD: "The region of these grids is jointly defined by 500 samples drawn from P and Q."

---

> > > > > ### Author Response · Authors · 2020-11-24
> > > > > **Thanks for the clarifying the question**
> > > > >
> > > > >
> > > > > Thanks a lot for your clarification. We understand the question now :)
> > > > >
> > > > > We can confirm that the 500 samples are enough for defining the borderline of the grid region (the min and max values of each dimension). This is not only verified empirically by our experiments, but is also supported theoretically by the Dvoretzky–Kiefer–Wolfowitz inequality, which states the relationship between the sample size n and the estimation error of the empirical distribution function. When substituting n=500 into this inequality one would see the estimation error there is very small, which means that the min and max values of these 500 samples indeed well represent the borderlines of the marginal distributions, and hence well define the grid region.

---

> > > > > > ### Comment · AnonReviewer1 · 2020-11-24
> > > > > > **Thank you; updated review**
> > > > > >
> > > > > > Thanks for clarifying this issue!
> > > > > >
> > > > > > I have updated my review and score to acknowledge your replies

---

> > > > > > > ### Author Response · Authors · 2020-11-25
> > > > > > > **Thanks!**
> > > > > > >
> > > > > > >
> > > > > > > Many thanks for increasing the score! The discussion there was really helpful for improving our work. Any further discussion is welcome :)

---

### Official Review · AnonReviewer4 · 2020-10-28
**Interesting and technically sound study, but clarity and soundness of the claims could be improved**

**Rating:** 6
**Confidence:** 4

**Review:**

### Summary
This study proposes a new method for computing summary statistics in the context of likelihood-free inference. The authors pose the problem as finding the summary statistics function (in the form of a neural network) that maximises the mutual information between parameters and summary statistics. The study is technically sound and will be of interest to the likelihood-free community and inspire further method development. However, the manuscript could be improved in terms of clarity and soundness of the claims.


### Quality
The paper is technically sound, but some of the claims are not sufficiently backed by empirical evidence:

-in the main section of the paper, tables 1-3, the authors report a significant improvement in performance compared to previous methods (SRE and SNPE) (with the exception of the Gaussian copula case, where SNPE seems to have a slightly better performance). However, in the appendix, the differences in performance (using the MMD metric) between algorithms are much more nuanced and in fact, it is unclear what algorithm is the best. It would be important for the authors to discuss these nuanced results in the main section of the paper, and be careful not to oversell their approach;

-related to the above point, most results presented in the paper are averages across three independent runs for each algorithm. This is somewhat insufficient to assess the real difference in performance between algorithms, and increasing this to at least 10 rounds would be critical.


### Clarity

The manuscript is for the most part clearly written. However, it sometimes lacks enough information for an expert reader to follow all the steps to evaluate and reproduce the results. Some comments to improve clarity:

-when describing SNL in the main section of the manuscript, the authors write that after getting an estimate of the likelihood:
"We then obtain the r-th estimate of the posterior by Bayes rule". However, the posterior is obtained via MCMC and this should be made explicit for clarity. This is not described at all in the manuscript, and I would urge the authors to also include a few more details on the MCMC procedure in the supplement;

-what were the hyperparameters used for SRE and SNPE? Did the authors test different parameterisations of these for the comparison between algorithms?


### Originality

The novelty of the study resides in framing the problem of computing the approximate sufficient statistics (in the context of likelihood-free inference) as a maximisation of mutual information between the parameters and the summary statistics.


### Significance of the work

The results suggest that the developed approach is a promising step towards improving likelihood-free methodology, especially in settings with high-dimensional data.


### Minor comments

Regarding the choice of SNPE-B over SNPE-C, the authors write "We select to compare with SNPE-B (Lueckmann et al., 2017) rather than the more recent SNPE-C (Greenberg et al., 2019) due to its equivalence to SRE shown in (Durkan et al., 2020)." However, Durkan et al. 2020 only showed similarity between SRE and SNPE-C and not strict equivalence, so the authors should rephrase this.


### Some typos:

-instead of "for automatically construction", "for automatic construction";

-instead of "problem-specific design of S as future works", "problem-specific design of S as future work";

-instead of "Likelihood-targeting approaches has the advantage that it could [...], but relies", "...have the advantage that they could...but rely...";

-instead of "our method does increases", "our method does increase";

-instead of "methods augmented with the proposed statistic [...] outperforms", "methods augmented with the proposed statistic [...] outperform";

-instead of "In Figure 2.(b)", "In Figure 2 (b)";

-instead of "in Figure 2.(c)", "in Figure 2 (c)";

-instead of "discreterized", "discretized".

---

> ### Author Response · Authors · 2020-11-21
> **Thanks for your useful comments; we have clarified details in empirical evaluations and improved the clarity**
>
> Issues with empirical evaluation - (a) MMD: the results for MMD here are just for reference and we think the results for JSD are more reliable. This is because the calculation of MMD is indirect, i,e it needs to first sample from the posterior, then computes the discrepancy from these samples (by kernels). JSD, on the contrary, directly compares two analytical densities (i.e the true and learnt posteriors), which should be more accurate. The difference between the MMD and JSD results might also be due to the fact that JSD is more sensitive to small difference in two distributions (due to the $\log$ transformation). Nonetheless we will use more conservative language when comparing to other methods. (b) More runs: we will definitely re-run all the experiments with 10 times average as you suggested (but please note that this might not be very realistic within two weeks, but we will update it in the final version).
>
> Clarity - (a) MCMC in SNL: it is true that one can sample from the posterior in SNL by MCMC as done in the original paper. Here, to fairly compare between different LFI methods, we use simple rejection sampling in SNL (and also for SRE/SNPE) when sampling from the approximate posterior, so that each algorithm only differs in the way they learn the posterior. (b) Details for SRE/SNPE: we have now provided them in the appendix, where we carefully control that different LFI methods only differ in the way they learn the posterior.
>
> Originality - besides framing the problem of learning sufficient statistics in LFI as infomax learning, our work also provides a sufficient statistics perspective to rethink existing LFI methods (e.g SRE), as well as a new way to compare/select different statistics (e.g expert v.s learnt, by the proposed infomax criteria), or to even unify them (e.g expert $\cup$ learnt, by feeding them jointly to the $T$ network). These are not done in existing works. Finally, since summary statistics are used in practice in a wide range of LFI algorithms, more than we have compared to (for example in the synthetic likelihood approaches), we expect that our work would have a broad impact in likelihood-free inference.
>
> Minor comments (comments SNPE-B v.s SNPE-C) - thank you very much for pointing out this. We will be more careful on this point and revise it accordingly.
>
> Typos - we have fixed all the typos and will update in the next version. Thank you very much again for pointing out them.

---

### Author Response · Authors · 2021-03-15
**Camera-ready revision summary**


Below are the major changes in the camera ready version:

* **Density-free/ratio-free sufficient statistics learning**: as discussed in the rebuttal, our framework has the potential to learn sufficient statistics without estimating any density or density ratio. We have now provided formal evidence about this; please check the revised paper. This makes our method fundamentally different from ratio-estimating approaches, addressing the previous conceptual question raised by Reviewer 1 and 2.

* **Dimensionality of the sufficient statistics**: we have now also provided a simple-yet-effective principle for selecting the dimensionality of the sufficient statistics.

We sincerely thank the reviewers and the program chairs for the very enlightening discussions.

---

> ### Comment · AnonReviewer2 · 2021-03-15
> **Thans for the notifications!**
>
> Thanks for letting us now! I will have a look at it, it was pleasant reviewing your paper!

---

> > ### Author Response · Authors · 2021-03-16
> > **Thanks**
> >
> > You are welcome! Feel free to add any comments. Codes will be released very soon :)

---

### Decision · Program_Chairs · 2021-01-07
**Final Decision**

**Decision:**

Accept (Spotlight)

**Comment:**

This paper addresses a central problem in inference in implicit models-- classical approaches on such problems ('ABC') rely on computation of summary statistics, and multiple methods for automatically finding summary statistics have been proposed. This paper provides a fresh take on this classical problem, by providing a methods for finding information-maximising summary stats. The work is original, likely impactful, and carried out rigorously and carefully. The reviewers flagged some issues with empirical comparisons, as well as discussion or relevant work-- those issues mainly seem to have been resolved in the review process. Moreover, given the originality of the approach, and provided that the description of empirical comparisons and relationship with other work are carefully and conservatively worded, I believe this will be worth publishing even if it is not always the 'best' method on all problems.